# Intolerance of uncertainty does not significantly predict decisions about delayed, probabilistic rewards: A failure to replicate Luhmann, C. C., Ishida, K., & Hajcak, G. (2011)

Pedro L. Cobos[1,2]*, María J. Quintero[1,2], Francisco J. López[1,2], David Luque[1,2], Luis F. Ciria[3], y Joaquín Morís[1,2]*

1 Universidad de Málaga, Málaga, Spain, 2 Instituto de Investigación Biomédica de Málaga y Plataforma en Nanomedicina-IBIMA Plataforma BIONAND, Málaga, Spain, 3 Mind, Brain and Behavior Research Center (CIMCYC), Universidad de Granada, Granada, Spain

* jmoris@uma.es (YJM); p_cobos@uma.es (PLC)

This is a Registered Report and may have an associated publication; please check the article page on the journal site for any related articles.

## Abstract

Intolerance of Uncertainty (IU) is thought to lead to maladaptive behaviours and dysfunctional decision making, both in the clinical and healthy population. The seminal study reported by Luhmann and collaborators in 2011 showed that IU was negatively associated with choosing a delayed, but more certain and valuable, reward over choosing an immediate, but less certain and valuable, reward. These findings have been widely disseminated across the field of personality and individual differences because of their relevance to understand the role of IU in maladaptive behaviours in anxiety-related disorders. We conducted a study to replicate and extend Luhmann et al.'s results with a sample of 313 participants, which exceeded the size necessary (N = 266) to largely improve the statistical power of the original study by using the *small telescopes approach*. The results of our well powered study strongly suggest that the relationship between IU and the tendency to prefer an immediate, but less certain and less valuable reward is virtually negligible. Consequently, although this relationship cannot be definitely discarded, we conclude that it cannot be detected with Luhmann et al.'s (2011) decision-making task.

## Introduction

Intolerance of uncertainty (IU) has been defined as "an individual's dispositional incapacity to endure the aversive response triggered by the perceived absence of salient, key, or sufficient information, and sustained by the associated perception of uncertainty" [1]. A distinctive feature of IU is that uncertainty involves a future-oriented unpredictable component, a hallmark that makes a difference with intolerance of ambiguity, which would involve unpredictability regarding the "here and now" [2]. Unadjusted reactions to uncertain future relevant events have been claimed to be crucial to understand the relationship between IU and pathological anxiety, as this condition has been related to maladaptive anticipatory reactions to

**Data Availability Statement:** The behavioural data and questionnaire scores, completely anonymized, are available at the OSF repository at https://osf.io/qyk87/, with unrestricted access. The R script with all of the statistical analyses performed are available at https://osf.io/b8hfc/. Finally, the experimental task used is available at the OSF repository https://osf.io/qyk87/.

**Funding:** This research has been supported by grants PID2021-126767NB-I00 and PGC2018-096863-BI00 from the Spanish Ministry of Science, Innovation, and Universities (AEI/FEDER, UE), and grants UMA18-FEDERJA-051 and ProyExcel_00287 (Junta de Andalucía regional government). MJQ has been supported by a predoctoral grant from the Spanish Ministry of Science, Innovation, and Universities (FPU Programme, FPU18/00917). The funders had no role in study design, data collection and analysis, decision to publish, or preparation of the manuscript.

**Competing interests:** The authors have declared that no competing interests exist. Nor do any of the funding institutions have conflicting interests.

unpredictable future threats [3]. Consequently, it is not surprising that IU has been found to play a crucial role in several anxiety-related disorders [4–8], as well as a transdiagnostic vulnerability factor for the development and maintenance of anxiety and depression symptoms [7,9,10].

Studies about IU have greatly improved our knowledge about the concept itself, its assessment, its relationship with different mental disorders as well as with other dispositional factors [11]. However, there is insufficient knowledge about the expression of IU in terms of behaviour and decision-making [11]. Therefore, the assessment of IU has to rely mostly on self-report measures that may be more prone to subjective biases. Additionally, a better understanding about how IU relates to behaviour and decision-making is warranted if we are to find out the causal mechanisms through which IU promotes the development and maintenance of symptoms in different psychopathology conditions.

One extended idea about IU is that the incapacity to endure uncertainty in some individuals makes them engage in behaviours and decisions intended to turn uncertain situations into more predictable ones or to enhance perceived control [1,4,12,13]. Consistently, high IU individuals have been described as risk avoiders [1] who, if needed, may sometimes behave to gain a feeling of predictability even at the cost of efficiency [14–16]. However, Luhmann et al. [17] proposed that decision-making guided by IU may not always aim at avoiding risk or enhance perceived predictability. According to them, the motivation driving behaviour and decision-making in high IU people is the urgent need to escape from (or avoid) the distress caused by uncertainty. This hypothesis slightly differs from the previous one in that it explicitly states that the aversion to uncertainty-related distress is greater than the aversion to uncertainty itself. Interestingly, this idea has empirical implications. In some circumstances, high IU people may choose more uncertain outcomes of lower value to avoid, or escape from, longer periods of distress waiting for less uncertain outcomes of higher value.

Luhmann et al. [17] provided evidence supporting their hypothesis in an experiment with 50 non-clinical participants. They went through 100 trials in each of which they had to decide between selecting an immediate choice with a 50% chance of receiving 4 cents or waiting to select a delayed choice with a 70% chance of receiving 6 cents. In both cases, participants knew right after their response if they had obtained the reward or not. Luhmann et al. predicted that high IU participants, compared with low IU participants, would show a higher preference for the immediate but more uncertain and less valuable outcome over the delayed but less uncertain and more valuable outcome. Consistently, they found a negative association between IU scores and willingness to wait for the second stimulus, after controlling for trait anxiety (TA) and monetary delay discounting, as measured through Kirby and Marakovic's Delay-Discounting questionnaire [18].

Luhmann et al.'s [17] study has been cited very often, as their results have important implications about the role of IU in some psychopathologies, and the conception of IU. However, as far as we know, Tanovic et al.'s study [19] is the only attempt to replicate Luhmann et al.'s results. Unfortunately, both the analyses conducted and the results found by Tanovic et al. [19] were considerably different from Luhmann et al.'s. First, Tanovic et al. only found a relationship between inhibitory IU (I-IU, one of the two factors of the IU scale) and willingness to wait, whereas Luhmann et al. did not include the IU factors in the analyses. Second, Tanovic et al. did not conduct any analysis to assess the specificity of the relationship found between I-IU and willingness to wait. Finally, Tanovic et al. used a rather small sample size of 56 participants, which may explain the differences between their results and those found by Luhmann et al. To ascertain the true relationship between IU and willingness to wait under uncertainty, we decided to carry out a considerably strict replication and extension of Luhmann et al.'s study.

Our replication differed from the original study in a few respects. First, the sample size was larger than the original study to be able to detect a smaller effect. Second, the behavioural task was slightly modified to monitor participants' engagement and to have an additional control of their performance. As in the original study, we tested if the referred association is observed after controlling for trait anxiety. Another concern regarding the task used by Luhmann et al. [17] is that the ability to refrain from choosing immediate small rewards to get delayed and more valuable rewards may be related to impulsivity [20]. Therefore, as Luhmann et al. [17], we performed statistical analyses to test if the association between IU and willingness to wait is found after controlling for a delay discount factor calculated from participants' responses in a questionnaire based on decisions between monetary rewards differing in magnitude and delay. To extend the original findings, participants in our study fulfilled the Spanish version of the short UPPS-P impulsive behaviour scale (SUPPS-P) [21]. This way, we could assess the specificity of the relationship between IU and willingness to wait after controlling also for impulsivity, in addition to trait anxiety and delay discount.

## Method

Except for a few minor details referred below in the Method section, we stuck to the specifications made in our previously published registered report protocol [22].

### Participants

Initially, we planned to recruit participants from several Spanish universities [22], but finally, only participants from University of Málaga were recruited. 345 undergraduate students from the referred university participated in the experiment in exchange for a monetary reward and course credit. The amount of money earned depended on their performance during the decision-making task itself. We included participants with normal or corrected-to-normal vision. Recruitment started on 15 February 2022, and finished on 22 November 2022. Before being recruited, participants read and signed the informed consent by ticking an "Accept" box. Participants were naïve to the aim of the study to avoid expectation effects. 313 participants (254 females) from the initial sample were selected for data analyses as they met the selection criteria described in our previously published registered report protocol (see below in the Data selection section). The experimental procedure was approved by the Ethics Committee of University of Málaga (CEUMA-46-2020-H), complying with the Declaration of Helsinki [23]. The target sample size was calculated by executing an R [24] script (https://osf.io/va5db/) that uses the packages pwr (version 1.3–0) [25] and MBESS (version 4.6.0) [26,27]. Following the small-telescopes approach proposed by Simonsohn [28], we estimated a target effect size based on the original experiment. According to Simonsohn's proposal, the target effect size would be the effect size that can be detected with a power of 33% in the original study. In Luhmann et al.'s study, the main results were obtained from a multiple regression analysis where IU and TA scores, and the delay discount factor were included as predictors, and the percentage of trials in which a delayed choice was made, p(Wait), was included as the dependent variable. This analysis yielded significant regression coefficients for IU and delay discount, but not for TA. Based on this result, we considered two possible sample sizes to choose the most conservative one. In the original study, that multiple regression model had 3 degrees of freedom in the numerator and 45 in the denominator. For the omnibus regression, the effect size that could be detected with a 33% power in the original study is $f_2 = 0.081$. The sample size required to detect that effect size in a multiple regression with a power of 95%, using an $\alpha = .05$, would be of 215 participants. However, the main theoretical point is the relation between IU and the behavioural task, as measured by p(Wait). Focusing on the targeted correlation coefficient

between IU and p(Wait), the effect size that can be detected with a 33% power in the original study is $f_2 = 0.049$, and the sample size required to detect this effect with a power of 95% would be n = 266.

As mentioned before, we chose the most conservative sample size, n = 266. This sample-would be 5.4 times the original sample, and lead to a highly powered experiment. In our previous registered report [22], we stated that recruitment would stop as soon as a size of 266 participants were reached after excluding from the analyses the data from those participants who did not meet the selection criteria (see below). Instead of this procedure, we recruited a number of participants well over the target sample size to ensure that the final sample for data analyses have a size of 266 or greater.

## Questionnaires

Following Luhmann et al. [17], all participants completed the IU, TA, and delay discounting questionnaires. As discussed in the introduction section, an impulsivity test was also included. Specifically, we used the following questionnaires:

**The Spanish adaptation of the Intolerance of Uncertainty Scale.** The IUS [29,30] is a 27-item self-report measure that assesses the degree to which individuals find uncertainty to be distressing and undesirable (internal consistency of .91 and test-retest reliability of .78; [31]). The IUS includes two subscales known as Prospective Intolerance of Uncertainty (11-items) and Inhibitory Intolerance of Uncertainty (16-items). Items are rated on a five-point Likert scale ranging from 1 (*not at all characteristic of me*) to 5 (*extremely characteristic of me*).

**The Spanish adaptation of the trait subscale of the State Trait Anxiety Inventory, Form Y.** This subscale of the STAI [32,33] is a 20-item self-report questionnaire with good psychometric properties (internal consistency between .90 and .95, and test-retest reliability between .84 and .91). STAI includes two subscales known as Trait Anxiety (20 items) and State Anxiety (20 items), but participants only completed the Trait Anxiety (TA) subscale. Items in the TA subscale are rated on a four-point Likert scale ranging from 0 (*nothing*) to 3 (*a lot*).

**The Spanish adaptation of the Delay-Discounting Test.** This test [34,35] is a 27-item monetary-choice questionnaire asking for individual preferences between smaller, immediate rewards and larger, delayed rewards varying on their value and time to be delivered (test-retest reliability between .63 and .77). After reading each item (e.g., "Would you prefer 55€ today, or 75€ in 61 days?"), participants have to indicate which alternative they would prefer to receive by marking the alternative in the questionnaire.

**The Spanish adaptation of the SUPPS-P Impulsive Behavior Scale.** This test [21,36] is a 20-item inventory designed to measure five distinct personality facets of impulsive behaviour: Positive urgency, negative urgency, lack of perseverance, lack of premeditation, and sensation seeking (internal consistency between .61 and .81). Items are rated on a four-point Likert scale ranging from 1 (*strongly agree*) to 4 (*strongly disagree*). Previous studies have shown weak correlations between the Delay-Discounting Test and different trait measures of impulsivity [34,37]. This indicates that this test might be capturing only some aspects of impulsivity. Because of this, we expected that including SUPPS-P in the analysis, compared to using only the Delay-Discounting Test, would provide additional information, and allow a better interpretation of the results obtained (see below).

## Procedure

Although the study was run using Google Forms and a JavaScript-coded program that were designed to conduct the study online [22], we finally decided to use our laboratory for the sake

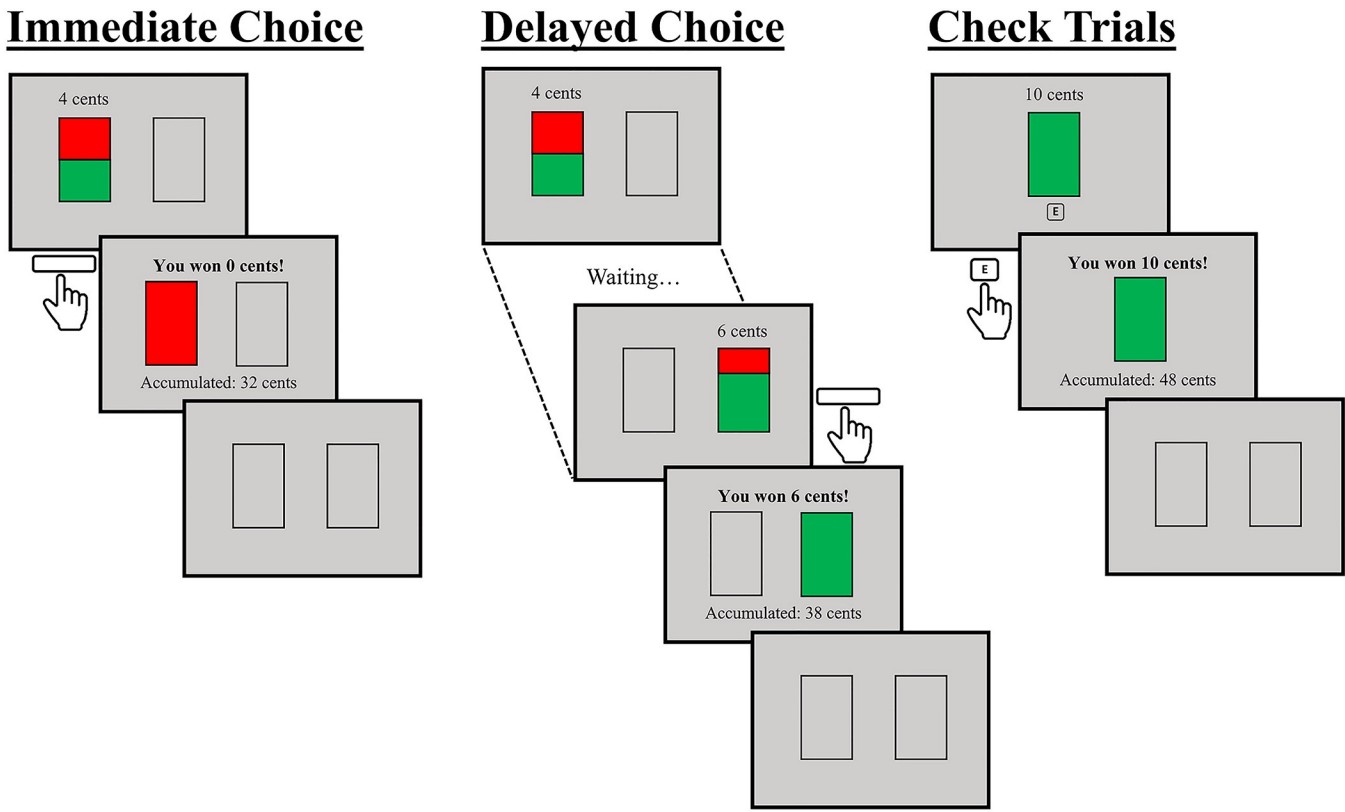

**Fig 1. Sequence of the decision-making task.** Left: Trial in which the immediate choice is selected, but no reward is won. Centre: Trial in which the delayed choice is selected, and the reward is won. Right: Check trial to monitor the participant's engagement, in which an appropriate response is provided, and the reward is won.

of comparability between our experiment and Luhmann et al.'s (2011). Our laboratory is equipped with IBM-compatible PCs and participants' responses were registered through a standard QWERTY keyboard. Participants entered our laboratory in groups of different sizes (up to a maximum of 20 per session) and read an informed consent that had to be signed to participate. Then they completed the series of questionnaires and performed a decision-making task. Questionnaires were completed online using Google Forms. Answering the whole set of items of each questionnaire was compulsory. The decision-making task was coded in JavaScript using Psychopy (version 2020.1) [38] and jsPsych (version 5.0.3) [39] and was hosted and deployed online on a secure server at University of Málaga. The task is available at the OSF repository https://osf.io/qyk87/.

After completing all the questionnaires, participants accessed the online platform to perform the decision-making task described in Luhmann et al.'s study. In this task, each trial began with the presentation of two empty rectangles displayed on a grey background, side by side at the centre of the computer screen, for a minimum period of time of 0.5 s (see below). Then, the left rectangle was filled in with two colours (red and green). The colours in the rectangles provided information about the probability of being and not being rewarded (see Fig 1), which is represented by the size of the green and red areas, respectively (e.g., if both areas had the same size, the likelihood of receiving the reward was 50%). Simultaneously, the monetary value of the reward was displayed above the rectangle. Selecting this first option (i.e., the immediate choice) always led to a 50% chance of receiving a 4 cents reward. Alternatively, participants could wait for the appearance of a delayed choice indicated by the display of the

green and red areas in the right rectangle and the offset of these colours in the left rectangle. This delayed choice always led to a 70% chance of receiving 6 cents. Immediately after participants select any choice by pressing the spacebar, the rectangle was completely filled in with only one colour for 1000 ms, green or red, indicating whether or not, respectively, the reward was gained. Participants also received additional information through text messages telling whether they received the reward or not, and the money accumulated so far. The duration of the feedback was 1 s. To prevent large deviations from the probabilities described to the participants, reward events followed a pseudorandom distribution. In each block of ten choices of the same type, the number of wins and losses was fixed (5 and 5 in the case of immediate choices, and 7 and 3 in the case of delayed choices, respectively), being order randomized within blocks.

We also added a new type of trial to enhance and assess the participants' engagement with the task. Ten check trials were included in which the only choice was a 100% chance of receiving 10 cents, which was indicated by the display of a new rectangle at the centre of the screen completely filled in with green colour. Participants were allowed to select this choice by pressing key "E" on their keyboard within three seconds after the rectangle onset. If this choice was not selected in time, or another key was pressed, a message was displayed telling participants that they missed the possibility of gaining 10 cents. Note that participants had to press key "E", instead of the spacebar (i.e., the selection key in normal trials), to prevent them from performing the task in an automatic or inattentive way. Check trials were scheduled to occur pseudorandomly so that only one appeared among the last 5 trials of each 10-trial block. This new type of trials constitutes a modification of the original task that was meant not only to enhance the engagement with the task, but also to discard from further analysis the results from those participants who were not adequately paying attention to the stimuli.

A critical detail of the task was that the delay between the onset of the immediate choice and the delayed choice varied between 5 and 20 seconds according to a truncated exponential distribution to maximise uncertainty about the time for the second choice to appear. Crucially, participants were explicitly told that they could not move on to the next trial any sooner by selecting the immediate choice since this action would simply extend the following inter-trial interval (ITI). The ITI followed the same variable time schedule as the delays between choice options. Thus, if in a given trial the programmed delay between choice options was, for instance, 11 s, and the participant took 2 s to select the first choice, the next ITI lasted for 9.5 s, i.e., the result of adding the 11 s of programmed delay minus the 2 s response time to the minimum ITI duration of 0.5 s. Participants completed 10 practice trials to familiarize with the task procedure. In these trials, they were instructed to make specific choices (i.e., on half of the practice trials, they were requested to select the immediate choice, while on the other half, they had to select the delayed choice) to ensure that they were exposed to the full range of possible outcomes. Two additional practice trials were also added to get participants familiar with the check trials. In both of them, they were instructed to press key "E" within three seconds. After practice trials, a total of 100 trials (plus 10 check trials) were administered. The task lasted for, approximately, 25 minutes.

## Data selection

As planned in our registered report protocol [22], participants not completing the entire session and the questionnaires were excluded from analyses (n = 3). In addition, participants responding incorrectly on more than two check trials (n = 21), or with more than 10% of reaction times (RTs) below 200 ms when choosing the immediate choice (n = 0), or more than 10% of RTs greater than 3000 ms when choosing the delayed choice (n = 8), or more than 20

responses to space bar or key "E" during ITI (n = 1), were excluded from analyses. Note that one of the participants met more than one exclusion criterion. As a result, the data from 32 participants were removed from the analyses, leaving a final sample of 313 participants [254 females; $M_{age}$ = 19.55 ($Min_{age}$ = 17, $Max_{age}$ = 59)].

## Statistical analysis

The behavioural data and questionnaire scores, completely anonymized, are available at the OSF repository at https://osf.io/qyk87/, with unrestricted access. The analyses performed were straightforwardly derived from Luhmann et al. (2011). The R script with all of the analyses described in this section are available at https://osf.io/b8hfc/. The packages lm.beta (version 1.7–2) [40], (version 0.1.1) [41], patchwork (version 1.1.2) [42] and tidyverse (version 2.0.0) [43] were used in this script.

We calculated the descriptives and zero-order correlations of the variables probability of waiting in the behavioural task [p(Wait)], IUS score, TA score, and discount factor, i.e. the natural logarithm of the k parameter in the delay discount factor used by Luhmann et al. (1/1+k), Delay-Discount. We used this logarithmic transformation after observing that the parameter distribution was heavily skewed. The transformation ensured a correct fit of the linear models, as shown by the diagnostic plots mentioned below. We also included SUPPS-P scores from the Impulsive Behaviour Scale in the referred analysis. As in the case of Luhmann et al., the main analysis consisted of a hierarchical linear regression (see Table 1). Using p(Wait) as the dependent variable, two models were considered: The first one included TA and Delay-Discount as predictors (Model 1), and the second model included IU as an additional predictor (Model 2). Extending Luhmann et al.'s study, an additional hierarchical linear regression analysis was planned. Model 1, that has TA and Delay-Discount as predictors was compared with a model that included SUPPS-P as an additional predictor (Model 3), and this model, in turn, was compared with a final model that also included IU (Model 4). In both hierarchical regression analyses, the difference between the models were tested using an ANOVA.

The comparison between Model 1 and Model 2 would indicate if there is a relationship between IU and p(Wait) after controlling for TA and Delay-Discount scores. This is the same comparison that was carried out by Luhmann et al. (2011), who found a significant positive association between IU and p(Wait). The comparison between Model 1 and Model 3 tested a possible relationship between SUPPS-P and p(Wait) after controlling for TA and Delay-Discount scores. Finally, the comparison between Model 3 and Model 4 assessed the relationship between IU and p(Wait) after controlling, additionally, for SUPPS-P. These two final comparisons are an extension of Luhmann et al.'s study, as described before.

As in Luhmann et al., the association between the same predictors and the median reaction time in trials with an immediate choice was tested with the same hierarchical regression analyses just described using the median response time in trials with an immediate choice as the dependent variable. This analysis would provide an additional chance of finding a relationship between IU and avoidance of waiting for the delayed choice. For each participant, the median

**Table 1. Summary of the regression models.**

| Models | Predictors | Compared with |
|---|---|---|
| Model 1 | TA, Delay-Discount | |
| Model 2 | TA, Delay-Discount, IU | Model 1 |
| Model 3 | TA, Delay-Discount, SUPPS-P | Model 1 |
| Model 4 | TA, Delay-Discount, SUPPS-P, IU | Model 3 |

response time was calculated using the reaction time of all the immediate choice trials of the original condition of the behavioural task (i.e., excluding the check trials).

Two diagnostic plots were also developed for each model: a scatterplot of the residuals and predicted values, and a Q-Q plot. If the diagnostic plots showed that the linear regression assumptions are not met, a robust linear regression technique (MM-estimates, as implemented by the MASS package in R) was planned to be used. After performing the logarithmic transformation of the delay discount parameter (see footnote 1) the plots showed a good fit of the models and a standard linear regression was used.

Two variables were calculated considering the proportion of delayed choices. The first one was the proportion of delayed choices when the previous trial was a nonreinforced, delayed-choice trial. The second one was the proportion of delayed choices after any other type of standard trial (i.e., trials after a checking trial were ignored). As in Luhmann et al., a paired *t*-test between these two variables was calculated, as well as the Pearson correlation between the difference of these two proportions and the participants' scores in each questionnaire.

### Exploratory analysis

Exploratory analyses [44,45] were carried out to separately study the role of the different factors of the SUPPS-P scale [46], as well as the prospective and the inhibitory factors of the IU scale [47]. We did not have any specific hypothesis regarding these analyses.

## Results

Table 2 shows the mean, median, standard deviation, and range of all the dependent variables analysed in this study, and Table 3 shows the Pearson correlation coefficients between IU, TA, Delay-Discount, SUPPS-P, and p(Wait).

### Relationship between IU and p(Wait)

We started by performing a regression analysis taking p(Wait) as the dependent variable and the predictors comprised in Model 1 (TA and Delay-Discount). The results revealed that the variance explained by the model was significant, $[R^2 = .07, F(2, 310) = 10.81, p < .001]$, as well as the regression coefficient of Delay-Discount, $[\beta = -.25, t(310) = -4.64, p < .001]$. The

**Table 2. Mean, median, standard deviation, and range of all the variables analysed in the present study.**

|                | Mean | Median | SD   | Range         |
|----------------|------|--------|------|---------------|
| IU             | 71.7 | 71     | 19.0 | [31, 121]     |
| TA             | 49.6 | 49     | 11.2 | [24, 76]      |
| Delay-Discount | -5.5 | -5.6   | 1.39 | [-8.8, -1.4]  |
| SUPPS-P        | 46.1 | 46     | 8.8  | [25, 71]      |
| p(Wait)        | .69  | .71    | 0.25 | [0, 1]        |
| Median RT Im   | 1.4  | 1.12   | 1.15 | [0.32, 10.3]  |
| p(WaitLoss)    | .57  | .62    | 0.36 | [0, 1]        |
| p(Waitno Loss) | .72  | .76    | 0.24 | [0, 1]        |

*Note*: IU = intolerance of uncertainty, TA = trait anxiety, SUPPS-P = Short UPPS-P Impulsive Behaviour Scale, p(Wait) = probability of selecting the delayed choice, Median RT Im = Median of response times in trials in which the immediate choice was selected, p(WaitLoss) = probability of selecting the delayed choice after losing the reward when the delayed choice was selected in the previous trial, p(Waitno Loss) = probability of selecting the delayed choice conditional upon any other possibility.

**Table 3. Bivariate Pearson correlation coefficients between the main measures of the study.**

|                | IU      | TA      | Delay-Discount | SUPPS-P |
|----------------|---------|---------|----------------|---------|
| TA             | .76**   |         |                |         |
| Delay-Discount | .03     | .00     |                |         |
| SUPPS-P        | .13*    | .24**   | .00            |         |
| p(Wait)        | .01     | .01     | -.25**         | .08     |

*Note*: * stands for significance below .05; ** stands for significance below .001.

regression coefficient of TA was not significant ($\beta$ = .01, $p$ = .786). The analysis based on Model 2, which added IU to Model 1, yielded almost identical results regarding Delay-Discount [$\beta$ = -.26, $t(309)$ = 4.63, $p <$ .001], and the non-significant regression coefficients of TA [$\beta$ = .01, $t(309)$ = 0.11, $p$ = .91] and IU [$\beta$ = .01, $t(309)$ = 0.09, $p$ = .932]. Consistently, Model 2 did not improve significantly the variance explained by Model 1 [$\Delta R^2 <$ .001, $F(1, 309)$ = 0.01, $p$ = .932]. Finally, we added SUPPS-P to Model 1 to assess the association between impulsivity and p(Wait) (Model 3) finding almost identical results regarding Delay-Discount [$\beta$ = -.26, $t(309)$ = 4.65, $p <$ .001] and the non-significant regression coefficients of TA [$\beta$ = -.004, $t(309)$ = 0.08, $p$ = .938] and SUPPS-P [$\beta$ = .082, $t(309)$ = 1.46, $p$ = .146]. Consistently, Model 3 did not significantly improve the variance explained by Model 1 [$\Delta R^2$ = .006, $F(1, 309)$ = 2.12, $p$ = .146]. Following the results reported above concerning the association between IU and p(Wait), the comparison between Model 3 and Model 4 was not significant [$\Delta R^2 <$ .001, $F(1, 309)$ = 0.04, $p$ = .85].

## Relationship between IU and median RTs in immediate choice trials

The same regression analyses considered in the previous section were conducted taking the participants' median response times in immediate choice trials as the dependent variable. None of the models or their coefficients were significant. The analysis based on Model 1 yielded non-significant regression coefficients for Delay-Discount [$\beta$ = -.025, $t(277)$ = 0.42, $p$ = .677] and TA [$\beta$ = -.077, $t(277)$ = 1.28, $p$ = .201], as well as a non-significant $R^2$ [$R^2$ = .007, $F(2, 277)$ = 0.91, $p$ = .405). Adding IU (Model 2) did not significantly improve the variance explained [$\Delta R^2 <$ .001, $F(1, 276)$ = 0.05, $p$ = .822], and revealed a non-significant regression coefficient for IU [$\beta$ = .02, $t(276)$ = 0.22, $p$ = .822]. Finally, adding SUPSS-P to Model 1, did not either improve the variance explained [$\Delta R^2$ = .004, $F(1, 276)$ = 1.14, $p$ = .286], nor did it reveal any significant regression coefficient for SUPPS-P [$\beta$ = -.06, $t(276)$ = 1.05, $p$ = .295]. Equivalent results were found in the case of Model 4 [IU $\beta$ = .01, $t(276)$ = 0.12, $p$ = .905), and its comparison with Model 3 [$\Delta R^2 <$ .001, $F(1, 275)$ = 0.01, $p$ = .905].

## Relationship between IU and outcome sensitivity

Instead of calculating outcome sensitivity as p(WaitLoss after delayed choice)—p(Waitno Loss after delay choice) (see Luhmann et al., 2011), we calculated p(Waitno Loss after delay choice)—p(WaitLoss after delay choice) (i.e., the probability of selecting the delayed choice after losing the reward when the delayed choice was selected in the previous trial was subtracted to the probability of selecting the delayed choice conditional upon any other possible outcome and choice selection in the previous trial). This change was meant to test for a positive correlation between self-report measures and outcome sensitivity, which could be interpreted in a more intuitive way than a negative correlation. The analyses of the correlations planned in our previous registered report yielded the significant correlation between Delay-Discount and outcome

sensitivity ($r = .2$, $p < .001$), and the non-significant correlation between outcome sensitivity and IU ($r = .01$, $p = .811$), TA ($r = -.03$, $p = .602$) and SUPPS-P ($r = .003$, $p = .946$). Consequently, the participants scoring high, compared with low, on Delay-Discount (those for whom monetary reward tended to lose value more quickly as delay time increases) were more likely to select the immediate choice after losing the delayed reward in the previous trial.

## Exploratory analyses

As explained in our previously published registered report protocol [22], we conducted exploratory analyses to assess the relationship between our main dependent measures [p(Wait), median RTs in immediate choice, and outcome sensitivity] and the two subscales of IU (prospective intolerance of uncertainty, P-IU, and inhibitory intolerance of uncertainty, I-IU) and the five subscales of SUPPS-P (negative urgency, positive urgency, lack of premeditation, lack of perseverance, and sensation seeking). Regarding p(Wait), we only found a significant correlation with lack of perseverance, $r = .13$, $t(311) = 2.25$, $p = .025$ (largest correlation with the remaining subscales = -.075). However, this correlation became non-significant when using the Bonferroni correction to protect against Type-I error. As for median RTs in immediate choice and outcome sensitivity, we did not find any significant correlation (largest $r = -.11$, smallest $p = .064$).

## Discussion

IU has been widely postulated in the literature as a main source of maladaptive and inefficient behaviour and decision making that may severely affect people suffering from anxiety-related mental disorders, but this claim lacks substantial empirical support [1]. As far as we know, Luhmann et al.'s study [17] provides the clearest evidence showing that people scoring high on IU tend to choose options that are riskier and less valuable in exchange for less time waiting under uncertainty. This strongly suggests that many examples of costly behaviour in anxiety-related psychopathologies, such as excessive avoidance, may be understood as instances of decisions aimed to avoid time enduring uncertainty [see also 12,13]. Given the relevance of this claim and the potential of Luhmann et al.'s results to lead future research, we decided to replicate and extend their study by conducting a highly powered experiment whose actual sample size (N = 313) was 6.3 times the size of the original study. Although our study was designed to detect an effect size 3 times smaller than the effect found in the original study with a statistical power above 95%, we failed to replicate Luhmann et al.'s (2011) results concerning the role of IU. This vulnerability trait for anxiety-related disorders was not found to be significantly associated with the probability of selecting the delayed choice, the time spent before selecting the immediate choice, or with outcome sensitivity. Moreover, all the correlations and standardised regression coefficients found for IU in all the analyses were negligible (greatest value = .02). Consequently, our results strongly suggest that IU is far from playing a significant role in choosing between the immediate and delayed rewards in Luhmann et al.'s task. In other words, we could not find convincing evidence supporting the claim that the increase in IU is associated with more costly responses, i.e., less valuable and less certain rewards, in exchange for less time waiting under uncertainty.

The only factor that was found to play a significant role in our study was the Delay-Discount factor. Specifically, we found a significant negative association between Delay-Discount and p(Wait), and a significant positive association between Delay-Discount and outcome sensitivity. Therefore, those participants who tend to devalue rewards more quickly as they are delayed had a greater tendency to select the immediate-reward choice, and to switch from a delayed choice to an immediate choice selection if the delayed choice selection was

unrewarded in the previous trial. This result is interesting in itself because it indicates that Luhmann et al.'s decision making task is suitable as a tool to detect the behavioural expression of individual differences in delay discounting. At first glance one may think that this result is not surprising given that participants are asked to choose between an immediate, less valuable reward, and a delayed, more valuable reward. However, the sensitivity of the task performance to delay discounting may seem less obvious if we consider that participants only have to wait for a few seconds to select the more valuable choice, and that selecting the immediate choice does not have any effect on the amount of time waiting for the next selection. It only shifts the waiting time from the inter-stimulus interval to the inter-trial interval.

A possible explanation for our failure to replicate Luhmann et al.'s (2011) results could be related to homogeneity regarding IU scores. Given the much larger sample size used in our study, we did not expect to find less variability in our data than in Luhmann et al.'s (2011) study. In fact, the standard deviation found in our study ($SD_{IU}$ = 19, $Range_{IU}$ = [31, 121]) was larger than that found in Luhmann et al. ($SD_{IU}$ = 14.32). The difference in mean IU between our study and Luhmann et al.'s could also potentially explain the different results found. Participants in our study scored higher on IU ($M_{IU}$ = 71.7) than Luhmann et al.'s (2011) participants ($M_{IU}$ = 61.12). According to Luhmann et al.'s hypothesis, higher scores on IU should lead to lower scores in p(Wait). In an extreme case, having a sample with very low p(Wait) scores may hinder the finding of a relationship between IU and p(Wait) as a result of a sort of floor effect. However, contrary to what Luhmann et al.'s hypothesis predicts, the mean p(Wait) found in our study ($M$ = .69) was, if any, higher than in the original study ($M$ = .6). Moreover, p(Wait) was found to significantly correlate with Delay-Discount, which debunks any argument based on lack of sensitivity of p(Wait) to explain the absence of a significant association with IU.

Another feature of our sample that deserves some consideration is the unbalanced proportion of males (18.8%) and females (81.2%). Interestingly, exploratory analyses revealed that male participants tended to wait for the more valuable and likely option ($M_{p(Wait)}$ = .8) substantially more than female participants ($M_{p(Wait)}$ = .66), $t(311)$ = 4.1, $p < .001$, $d$ = 0.6, although the differences between males and females in IU ($M$ = 70.14 vs $M$ = 72.06, respectively) and Delay-Discount ($M$ = -5.51 vs $M$ = -5.54, respectively) were not significant (smallest $p$ = .48). Unfortunately, it is difficult to assess the relevance of the proportion of males and females to explain the differences between our results and those by Luhmann et al. (2011) because they did not report such demographic data. Assuming that their sample was rather balanced, our results suggest that the high proportion of females in our sample could not explain the higher mean p(Wait) found in our study compared with Luhmann et al.'s study. At the same time, our results also suggest that whatever may be the reason for the impact of sex on p(Wait), it seems that it is related neither to IU nor to Delay-Discount. Therefore, although the difference in p(Wait) between males and females may be interesting in itself and may deserve future research, it is very unlikely to be the cause of the conflict between our results and those by Luhmann et al. (2011).

A final consideration relates to the period in which the study was carried out. The experiment started near after the lift of COVID-19 restrictions such as the lockdown. Although the participants have been recruited until November of 2022, many of them answered the questionnaires and performed the decision-making task in early 2022. According to recently published papers [48–51], the exacerbation of intolerance of uncertainty during the pandemic has played a role in the increase of the presence and intensity of a good number of mental-disorder symptoms, including symptoms of anxiety-related mental disorders. In fact, this circumstance may explain the high scores on IU and TA ($M_{TA}$ = 49.56) in our sample compared with Luhmann et al. (2011) ($M_{TA}$ = 43.08). Although this particularity of our study has to be taken into

account, the considerations made above regarding the role of differences in IU between our sample and the sample in Luhmann et al.'s study lead us to cast serious doubts on the possible impact of the pandemic situation on our results.

To conclude, our results suggest that the task designed by Luhmann et al. may not be suitable for detecting a relationship between IU and the preference for immediate, less certain, and less valuable rewards over delayed, more certain, and more valuable rewards. This does not mean that Luhmann et al.'s hypothesis that people with high IU prefer to take costly actions to avoid waiting under uncertainty is necessarily wrong. We think that this hypothesis deserves more chances to find empirical support as it fits our daily-life experience regarding the effects of uncertainty. Students waiting for exam results, for instance, frequently report a strong desire for receiving immediate information even if it is to be informed that they failed the exam. However, even if Luhmann et al.'s hypothesis is correct, their decision-making task has to be improved to provide evidence supporting it.

## Author Contributions

**Conceptualization:** Pedro L. Cobos, y Joaquín Morís.

**Data curation:** María J. Quintero, y Joaquín Morís.

**Formal analysis:** y Joaquín Morís.

**Funding acquisition:** Pedro L. Cobos, Francisco J. López, David Luque.

**Investigation:** Pedro L. Cobos, María J. Quintero, Francisco J. López, y Joaquín Morís.

**Methodology:** Pedro L. Cobos, Luis F. Ciria, y Joaquín Morís.

**Resources:** Pedro L. Cobos, Francisco J. López, y Joaquín Morís.

**Software:** y Joaquín Morís.

**Supervision:** Pedro L. Cobos, María J. Quintero, Francisco J. López, David Luque, y Joaquín Morís.

**Writing – original draft:** Pedro L. Cobos, Luis F. Ciria, y Joaquín Morís.

**Writing – review & editing:** Pedro L. Cobos, Francisco J. López, David Luque, y Joaquín Morís.

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
