## [Decision Letter · Decision Letter 0]

24 Jun 2024

Intolerance of uncertainty does not significantly predict decisions about delayed, probabilistic rewards: a failure to replicate Luhmann, C. C., Ishida, K., & Hajcak, G. (2011)

PONE-D-24-02600

Dear Dr. Cobos Cano,

We’re pleased to inform you that your manuscript has been judged scientifically suitable for publication and will be formally accepted for publication once it meets all outstanding technical requirements.

Kind regards,

Rei Akaishi

Academic Editor

PLOS ONE

   "This research has been supported by grants PID2021-126767NB-I00 and PGC2018-096863-BI00 from the Spanish Ministry of Science, Innovation, and Universities (AEI/FEDER, UE), and grants UMA18-FEDERJA-051 and ProyExcel_00287 (Junta de Andalucía regional government). MJQ has been supported by a predoctoral grant from the Spanish Ministry of Science, Innovation, and Universities (FPU Programme, FPU18/00917)"

Please respond by return e-mail so that we can amend your financial disclosure and competing interests on your behalf.

Additional Editor Comments (optional):

The reviewer suggested acceptance.

Reviewers' comments:

Reviewer's Responses to Questions

**Comments to the Author**

1. Does the manuscript adhere to the experimental procedures and analyses described in the Registered Report Protocol?

If the manuscript reports any deviations from the planned experimental procedures and analyses, those must be reasonable and adequately justified.

Reviewer #1: Yes

2. If the manuscript reports exploratory analyses or experimental procedures not outlined in the original Registered Report Protocol, are these reasonable, justified and methodologically sound?

A Registered Report may include valid exploratory analyses not previously outlined in the Registered Report Protocol, as long as they are described as such.

Reviewer #1: Yes

3. Are the conclusions supported by the data and do they address the research question presented in the Registered Report Protocol?

The manuscript must describe a technically sound piece of scientific research with data that supports the conclusions. The conclusions must be drawn appropriately based on the research question(s) outlined in the Registered Report Protocol and on the data presented.

Reviewer #1: Yes

4. Have the authors made all data underlying the findings in their manuscript fully available?

Reviewer #1: Yes

5. Is the manuscript presented in an intelligible fashion and written in standard English?

Reviewer #1: Yes

6. Review Comments to the Author

Please use the space provided to explain your answers to the questions above. (Please upload your review as an attachment if it exceeds 20,000 characters)

Reviewer #1: I was not an original reviewer of the initial RR submission. The researchers did a good job carrying out the proposed plan, and indicating when deviations occurred. The data is available, results are transparent, exploratory analyses clearly labeled, and the conclusions reasonable.

7. PLOS authors have the option to publish the peer review history of their article (what does this mean?). If published, this will include your full peer review and any attached files.

Reviewer #1: No

---

## [Editor Report · Acceptance letter]

2 Jul 2024

PONE-D-24-02600 

PLOS ONE

Dear Dr. Cobos, 

I'm pleased to inform you that your manuscript has been deemed suitable for publication in PLOS ONE. Congratulations! Your manuscript is now being handed over to our production team.

Kind regards, 

on behalf of

Dr. Rei Akaishi 

Academic Editor

PLOS ONE